# Validation and Application of MD Anderson Symptom Inventory Module for Patients with Bladder Cancer in the Perioperative Setting

**DOI:** 10.3390/cancers14163896

**Published:** 2022-08-12

**Authors:** Mona Kamal, Neema Navai, Kelly K. Bree, Loretta A. Williams, Charles S. Cleeland, Shu-En Shen, Xin Shelley Wang

**Affiliations:** 1Department of Symptom Research, The University of Texas MD Anderson Cancer Center, 1515 Holcombe Boulevard, Unit 1450, Houston, TX 77030, USA; 2Department of Urology, The University of Texas MD Anderson Cancer Center, 1515 Holcombe Boulevard, Unit 1450, Houston, TX 77030, USA

**Keywords:** symptoms, patient-reported outcomes (PROs), validation, perioperative care, MD Anderson Symptom Inventory (MDASI), cystectomy, recovery

## Abstract

**Simple Summary:**

Patients’ perspectives should be incorporated into the perioperative care plan in patients with bladder cancer undergoing cystectomy. Our approach incorporated both patients’ perspectives and clinicians’ points of view to develop and validate the MDASI-PeriOp-BLC module. The MDASI-PeriOp-BLC module was found to be a valid, reliable, and concise tool to assess symptom burden and functional recovery in this specific cohort of patients.

**Abstract:**

Objectives: We developed and validated a disease-specific tool for perioperative patient-reported outcomes assessment for bladder cancer (BLC) patients undergoing radical cystectomy, The MD Anderson Symptom Inventory (the MDASI-PeriOp-BLC). Methods: Patients who underwent radical cystectomy were recruited. We used qualitative interviews and experts’ input to generate disease/treatment-specific items of the MDASI-PeriOp-BLC module; conducted item reduction; examined the psychometric properties of the resultant items for reliability, validity, and clinical interpretability; and conducted cognitive debriefing interviews to assess the tool’s performance. Results: A total of 150 BLC patients contributed to psychometric validation. We identified and defined eight BLC-specific module items (blood in urine, leaking urine, frequent urination, urinary urgency, burning with urination, constipation, changes in sexual function, and stomal problems). We included those 8 items in addition to 13 MDASI core symptoms and 6 interference items to form the MDASI-PeriOp-BLC module. Cronbach alphas were 0.89 and 0.90 for the 21 severity items and the 6 interference items, respectively. Test–retest reliability (intra-class correlation) was 0.92 for the 21 severity items. The MDASI-PeriOp-BLC module significantly differentiated the patients by performance status (*p* < 0.0001). Conclusions: The MDASI-PeriOp-BLC is a valid, reliable, and concise tool for monitoring symptom burden during perioperative care in BLC patients undergoing radical cystectomy.

## 1. Introduction

Bladder cancer (BLC) is the 10th most common cancer worldwide [1]. Many emerging treatment options for personalized management of BLC are available. Radical cystectomy has been the standard of care for treatment of non-metastatic muscle-invasive BLC and selected cases of non-muscle invasive disease [2,3,4]. Radical cystectomy is also a curative treatment option after neoadjuvant therapy for locally advanced stages and a palliative treatment option for metastatic disease [4,5,6,7,8].

In addition to the typical postoperative symptom burden experienced by many surgical patients [9], radical cystectomy represents a particularly morbid operation with unique recovery challenges. Contemporary series evaluating patients undergoing radical cystectomy demonstrate complications occurring in as many as 54–80% of patients during the first 90 days after surgery [10,11,12]. Furthermore, surgery and the subsequent recovery can negatively impact daily activities, social function [13,14], and body image [15]. Orthotopic urinary diversion carries an increased risk of urinary incontinence and retention, with additional symptom burden and detrimental effects on quality of life (QOL) [16,17].

To improve cancer patients’ experience, the U.S. Food and Drug Administration (FDA) and European Medicines Agency provide continuous guidance on the implementation of patient-reported outcomes (PROs) in clinical trials [18,19]. The most recent draft from the FDA emphasizes the importance of utilizing validated PROs as measurement tools, with specific considerations given to cancer clinical trials [20].

Within BLC, patient-reported symptoms and functional outcomes before and after radical cystectomy have been reported [14,21,22,23,24]; however, little progress has been made in terms of utilization, assessment, and understanding of PROs as outcomes in clinical trials for BLC [25,26]. PROs facilitate close monitoring of perioperative symptom burden and subsequently improve surgical outcomes, especially in the era of patient-centered care [27] and the Enhanced Recovery After Surgery (ERAS) protocols [28]. Monitoring the perioperative symptom burden and functional outcomes might accelerate postoperative recovery, increase patient satisfaction, and decrease postoperative complications. Therefore, to improve QOL during survivorship, we need to identify those patients at highest risk for severe symptoms and/or complications, aiming to optimize their recovery with proactive symptom-reduction and rehabilitation plans [29,30,31,32,33]. Thus, better understanding and optimized implementation of PROs are critically needed in the care of patients with BLC [34]. Furthermore, it has been proven that considering patient perspectives is an essential component of patient-centered care in genitourinary settings, in general. For example, the functional and psychological aspects may influence the treatment choice and potentially the outcomes in patients with prostate cancer [35].

To facilitate further implementation of PROs in routine practice [36], a validated PRO assessment tool is needed to address patients’ needs and track functional outcomes after radical cystectomy, especially among elderly patients with comorbid medical conditions, who represent the vast majority of BLC patients [37].

The primary objective of this study was to develop and psychometrically validate a perioperative version of the MD Anderson Symptom Inventory (MDASI), a well-established, comprehensive PRO assessment tool [38], for measuring symptom burden and functional recovery in patients undergoing radical cystectomy for BLC, termed the MDASI-PeriOp-BLC. We expect that using this tool will help monitor symptom burden during the perioperative period and that the resultant data will be incorporated into perioperative discussions and care to better meet patient care needs and improve education regarding postoperative expectations.

## 2. Materials and Methods

### 2.1. Participants

Patients treated with radical cystectomy at The University of Texas MD Anderson Cancer Center in Houston, Texas, between 16 September 2016 and 27 June 2019 were recruited. In this cross-sectional study, the initial and follow-up assessments were 1–2 days apart for test–retest purposes. This study was approved by the Institutional Review Board, and written informed consent was obtained from all patients. Eligible patients included those at least 18 years old who spoke English; had BLC and been scheduled for or had undergone radical cystectomy with curative or palliative intent; had no diagnosis of active psychosis or severe cognitive impairment; understood the study’s intent; and were willing to participate.

### 2.2. Data Collection

Eligible patients were enrolled at various time points: during the preoperative clinic visit, after the surgery but before discharge from inpatient service, or during the first follow-up (postsurgical) clinic visit at MD Anderson before receiving any other cancer treatment. Twenty patients from the validation study sample participated in the cognitive debriefing.

PRO measures were either completed online by the patients or over the phone by a trained study coordinator. The study coordinator assessed the patients’ performance status at the initial assessment using the Eastern Cooperative Oncology Group (ECOG) performance status scale [39], and clinical, demographic, and pathological data were collected via chart review. Data on the presence of comorbid conditions according to the Charlson Comorbidity Index [40] were also collected.

### 2.3. Development of the MDASI-PeriOp-BLC

Generation of candidate PeriOp-BLC symptom items (content validity) was followed by FDA guidance for development and validating PRO tools [18]. Patients who had undergone radical cystectomy for BLC within the last 3 months participated in semi-structured qualitative interviews about their experience. Using a standard interview guide, trained and experienced research staff in qualitative interviewing asked patients to describe their experience with BLC and radical cystectomy. Probe questions were asked as needed to elucidate the patient’s symptom experience. Interviews were digitally audio-recorded and transcribed verbatim by professional transcriptionists. Interviewers verified the accuracy of the transcriptions. Field notes on the circumstances of each interview were dictated by the interviewers following the interview and transcribed.

An experienced qualitative researcher (LAW) analyzed the transcripts using descriptive exploratory techniques. Symptoms that patients reported only before the surgery (disease-related symptoms) and symptoms that occurred after the surgery (peri-op-related symptoms) were identified. Symptoms were named based on patients’ descriptions of the symptoms. Interviews were continued until saturation was reached. Saturation was defined as no new symptoms identified in three consecutive interviews. A final list of symptoms with the number of patients identifying each symptom in an interview was compiled for consideration by an expert panel.

The cognitive debriefing process was another step utilized for the development of the MDASI-PeriOp-BLC. Twenty patients participated in an interview-based cognitive debriefing. The debriefing assessed ease of completion, comprehensibility, acceptability, redundancy, use of the scoring system, item clarification, and content domain confirmation of the new instrument.

### 2.4. Psychometric Validation Process and Analysis

We calculated two measures of reliability: internal consistency reliability (all assessment time points) and test–retest reliability (two assessments performed 1–2 days apart). The test–retest pairs were administered pre-surgery or >4 days post-surgery, excluding the immediate postoperative period, as an assumption of test–retest reliability is the stability of the construct, and symptoms change rapidly in the first few days following surgery. Test–retest reliability was measured by the intra-class correlation coefficient with an acceptable level of 0.70.

To demonstrate convergent validity, we concurrently administered the European Organization for Research and Treatment of Cancer Quality of Life Core Questionnaire for muscle-invasive BLC (EORTC-BLM30) [41], a single-item assessment of overall QOL (SIQOL) [42], and the Functional Assessment of Cancer Therapy–Bladder Cancer–Additional Items (FACT-BL) [43], a validated symptom assessment scale adapted for patients with BLC. We assessed the correlation between MDASI items (MDASI-BLC symptom burden subscales (core and newly developed module-specific items); MDASI-WAW items (work, activity, and walking); MDASI-REM items (relations with other people, enjoyment of life, and mood)) and EORTC-BLM30 items [41]; SIQOL [42]; and FACT-BL [43]. The calculated sufficient sample size to evaluate and validate the MDASI-PeriOp-BLC was based on the ability of the instrument to distinguish between patients with poor and good performance status as a measure of known-group validity [38].

### 2.5. Applicability of Newly Developed Tool in the Delivery of Perioperative Care

We assessed patient compliance (percentage of patients completing each item). We also compared the symptom severity profiles across different time points (pre-surgery, during hospitalization (within 4 days after surgery), and postoperative (>4 days after surgery)).

In summary, we followed the following steps: (1) we conducted interviewers with the patients to get their experience with BLC and radical cystectomy; (2) the interviews were recorded, transcribed verbatim by professional transcriptionists, and analyzed; (3) we shared the final symptom list with an expert panel; (4) we performed a debriefing process to assess ease of completion, comprehensibility, acceptability, redundancy, use of the scoring system, item clarification, and content domain confirmation of the new instrument; (5) we assessed the reliability of the tool; (6) we assessed the validity of the tool by comparing the tool with the existing tools; (7) we assessed the applicability of the tool by assessing the patient compliance by calculating the percentage of patients who completed each item.

## 3. Results

### 3.1. Participants

A total of 150 patients were included in this validation study. Of those patients, 69 provided test–retest data. Demographic and disease-related characteristics are summarized in Table 1. Enrolled patients were predominantly elderly (mean age 67.7 years old, standard deviation [SD] (9.66)), men (81%), and white (94%). Most of the patients had an ECOG performance status of 0–1 (58%), and 42 (28%) had a comorbid condition. Furthermore, 87% percent had clinical stage 0-II disease, and 25% had non-muscle invasive bladder cancer with recurrent disease. The majority were treated with conventional urostomy (79%), followed by neobladder (19%) and continent urinary reservoir (3%) procedures. Seventy-nine percent of patients had received some type of prior anti-cancer treatment. Eighty-eight percent received care under an ERAS pathway [44].

### 3.2. Content Domain of the MDASI-PeriOp-BLC

#### Qualitative Interviews

Saturation was reached after 20 patients were interviewed about their disease and surgical experience. The mean age of the patients was 70 years (SD 8 years), with a range of 56 to 82 years. Sixteen participants (80%) were male, nineteen (95%) were white, and fifteen (75%) had undergone education beyond high school. The participants were an average of 25 days post-surgery (SD 25), with a range of 2 to 82 days. Thirty-five individual postoperative symptoms were mentioned in at least one interview. In addition, there were 16 individual symptoms mentioned in at least one interview that patients identified as occurring prior to surgery.

To determine the symptoms for the initial content domain for the module, we include the 13 symptoms from the MDASI core items as well as any additional symptoms that were identified in at least 20% of the qualitative interviews. Furthermore, any symptoms occurring prior to surgery must also have occurred postoperatively in order to be included. From the qualitative analysis, we found that six MDASI core symptoms (pain, fatigue, nausea, disturbed sleep, distress, and lack of appetite) were mentioned in 20% or more of the interviews. Five other MDASI core symptoms (problem with remembering, shortness of breath, numbness or tingling, drowsiness, and vomiting) were mentioned at least once in interviews as occurring following surgery. In addition, six other symptoms (constipation, abdominal bloating, diarrhea, discomfort, soreness, and leaking stoma) were mentioned as BLC peri-op-specific symptoms. The core item of pain, as well as the non-core items of burning with urination, blood in urine, urinary urgency, urinary frequency, and change in sexual function, were mentioned as BLC-related symptoms by ≥20% of the patients.

### 3.3. Expert Panel Review

The expert panel included patients (N = 11), caregivers (N = 10), nurses (N = 16), and physicians (N = 16). Overall, 65% (34/52), 65% (33/53), 58% (31/53), 53% (27/51), 50% (25/50), 68% (36/53), and 45% (24/53) of the panel rated difficulty urinating, leaking urine pain or burning with urination, frequent urination, urinary urgency, blood in urine, and changes in sexual functioning as very relevant items, respectively.

### 3.4. Final Item Determination

Although the item assessing changes in sexual function had a low response rate (data missing for 24% of respondents), the panel decided to keep this item because sexual dysfunction remains an important aspect of survivorship for many patients and this data could be useful for future patient education. Based on the qualitative interviews and expert panel discussions, 10 BLC-specific symptom items were generated and added to the original 13 MDASI core symptoms and 6 interference items to form the initial version of the MDASI-PeriOp-BLC. Two items (abdominal tightness and diarrhea) were dropped after running the bivariate analysis due to lack of sensitivity.

Bivariate analysis showed that increased scores for pain, fatigue, disturbed sleep, distress, poor appetite, drowsiness, and dry mouth were correlated with increased scores of interference items of general activity, difficulty walking, and mood changes (r > 0.4). High scores for nausea were correlated with increased scores of difficulty walking (r > 0.4). Increased sadness scores were moderately/highly correlated with increased scores of mood changes, distress, poor relations with others, enjoyment of life, and difficulty walking (r > 0.5). Even though urinary frequency was highly correlated with urinary urgency (r = 0.74), after consultation with the panel, we kept both of those items because they represent different clinical issues.

The final eight BLC-specific items added to the MDASI core items as revisional modules for psychometric validation were blood in urine, leaking urine, frequent urination, urinary urgency, constipation, burning with urination, changes in sexual function, and stomal problems.

### 3.5. Psychometric Validity

A total of 150 patients participated in the assessment of psychometric validity; 30, 52, and 65 patients contributed at 3 time points (pre-surgery, post-surgery ≤ 4 days, and post-surgery > 4 days).

### 3.6. Reliability

#### 3.6.1. Internal Consistency

The internal consistency was measured using Cronbach’s alpha, which is a coefficient that represents the degree of interrelationship between the items. Table 2 shows the observed internal consistency within the 13 core items, the 8 BLC-specific items, and the 6 interference items. The overall coefficient of the MDASI-PeriOp-BLC items was 0.726 for all items, and the recalculated coefficients after deletion of a single item at a time were similar to the overall coefficient, which indicates the importance of including each individual item and demonstrates the high reliability of the items. The coefficients were 0.89 for the 21 symptom severity items and 0.90 for the 6 interference items, showing the high internal consistency of the composite scales.

#### 3.6.2. Test–Retest Reliability

The intra-class correlation coefficient was 0.928 for the 13 core items, 0.922 for all 21 symptom severity items, and 0.909 for the 6 interference items. The consistent recalculated alpha coefficient after deletion of each item individually indicates the importance of including each item and demonstrates the high reliability of the items. The test–retest reliability remained excellent for module items analyzed during hospitalization (>4 days after surgery) (intra-class correlation coefficient = 0.832) and >5 days after radical cystectomy (intra-class correlation coefficient = 0.842).

### 3.7. Construct Validity

#### 3.7.1. Known-Group Validity

The known-group validity of the MDASI-PeriOp-BLC was supported by the ability of MDASI subscales (core, severity, WAW, REM) and BLC module items to differentiate the patients according to poor vs. good ECOG performance status (2–4 vs. 0–1, all *p* < 0.0001), as shown in Table 3A. Each of those MDASI subscales was able to significantly differentiate patients according to symptom severity profile at different periods (pre-surgery, post-surgery within 5 days, and post-surgery after 5 days; all *p* < 0.05), as shown in Table 3B.

#### 3.7.2. Convergent Validity

Spearman correlation analysis showed significant correlations between the FACT-Additional BL and the MDASI-interference (*r* = −0.45013), MDASI-PeriOp-BLC module (*r* = −0.35839), MDASI-severity (*r* = −0.41193), MDASI-WAW (*r* = −0.41167), and MDASI-REM (*r* = −0.44241) subscales, with all *p* < 0.0001, as indicated in Table 4. The MDASI-PeriOp-BLC module items were significantly correlated with the urinary symptoms and problems (*r* = 0. 49272), urostomy problems (*r* = 0. 34041), future perspective (*r* = 0. 2721), abdominal bloating and flatulence (*r* = 0. 26717), and body image (*r* = 0. 25759) items of the EORTC-BLM30 (all *p* < 0.05). SIQOL was significantly correlated with the core items (*r* = −0.39255), the BLC module items (*r* = −0.21439), symptom severity subscales (r = −0.39765), interference items (*r* = −0.48843), MDASI-WAW items (*r* = −0.43154), and MDASI-REM items (*r* = −0.51071) (all *p* < 0.05). The core, BLC-module, severity, interference, MDASI-WAW, and MDASI-REM items showed moderate correlation (*r* values of 0.4–0.5) with the abdominal bloating and flatulence and body image items of the EORTC-BLM30 (all *p* < 0.0001).

### 3.8. Cognitive Debriefing Results

Twenty patients participated in the cognitive debriefing. Overall, the MDASI-PeriOp-BLC was well received, easy to use and comprehend, and satisfactory. More than 60% of patients gave perfect scores to five of the seven cognitive debriefing categories, while 50% reported perfect scores on “redundancy”. All patients reported that the 0–10 scoring system was easy to use for rating their experiences and felt comfortable using it. Moreover, 95% of patients felt that the 0–10 scoring system was easy to understand. Furthermore, 60% of patients felt that the questions were not difficult to complete or difficult to understand, and 70% of patients felt very comfortable while answering the questions.

Three patients (15%; ages 54, 66, and 83 years) commented on the item assessing changes in sexual dysfunction. One patient described it as “difficult to understand”, while another patient said they “felt uncomfortable in answering”. The difficultly urinating item also was mentioned by three patients who called it “difficult to understand” or said they “felt uncomfortable in answering”, and three patients mentioned the sexual dysfunction item in the “redundancy” section. Adding an option of “N/A” or “some questions just felt does not apply” was mentioned five times throughout the questionnaire by four patients.

### 3.9. Application of the MDASI-PeriOp-BLC

#### 3.9.1. Patient Compliance

Apart from the “changes in sexual function” item, all MDASI-BLC items were completed by at least 87% of patients; 24% did not complete the “changes in sexual function” item. The rates of missing responses to changes in sexual function items from the EORTC-BLM30 were 6–70%. Also, 23–100% of men and 44–93% of women refused to answer one of the gender-exclusive questions.

#### 3.9.2. Symptom Severity

The most severe symptoms from MDASI-core were fatigue (mean 4.25), disturbed sleep (mean 3.6), pain (mean 3.3), drowsiness (mean 3.23), dry mouth (mean 2.85), and poor appetite (mean 2.85). The most severe BLC module symptoms included constipation (mean 2.36), changes in sexual function (mean 2.36), blood in urine (mean 1.5), frequent urination (mean 1.21), stomal problems (mean 1.07), and leaking urine (mean 1.05). Relevant core (pain, fatigue, nausea, sleep, remembering, appetite, drowsiness, and dry mouth), module-specific (blood in urine, frequent urination, burning with urination, and urgency), and interference (activity, mood, work, walking, and enjoyment of life) items significantly differed across the different time points (pre-surgery, within 4 days after surgery, and >4 days post-surgery) (Table 5). Overall, 53% of the patients reported at least one item as severe (score ≥ 7).

The scores of MDASI-core, MDASI-BLC module-specific, and MDASI-interference items did not significantly differ between patients on an ERAS pathway (N = 132, 88%) and those who were not.

### 3.10. Impact of Symptom Severity on Interference Items

Regression analysis showed that scores of some core items (pain, dry mouth, and sadness) were significantly associated with scores of the interference items (all *p* < 0.05). Among BLC module items, stomal problems, blood in urine, and burning with urination were significantly associated with interference items (all *p* < 0.05), while some core items (pain, poor appetite, and dry mouth) were significantly associated with the interference item of general activity (all *p* < 0.05).

## 4. Discussion

The MDASI-PeriOp-BLC represents a multiple-symptom assessment tool that was developed with item generation via integration of both patients’ and clinicians’ perspectives and psychometric validation. The MDASI-PeriOp-BLC represents a novel, validated, comprehensive PRO measurement tool that could be utilized in the perioperative care of BLC patients. After qualitative interviews, consultation with a diverse group of stakeholders on our expert panel, and psychometric validation analysis, along with MDASI-core, the resultant validated eight MDASI-PeriOp-BLC module items demonstrate satisfactory psychometric properties with regard to validity, reliability, and sufficient sensitivity; the items are: blood in urine, leaking urine, frequent urination, urinary urgency, constipation, burning with urination, changes in sexual function, and stomal problems. Cognitive debriefing was performed to identify difficult items or confusing questions, with patients reporting that the MDASI-PeriOp-BLC module was easy to use and easy to understand, and that the numerical scoring system was simple and straightforward.

As we strive to improve the care we are providing to BLC patients, the incorporation of appropriate perioperative counseling that addresses common patient questions/concerns and aids in identification of those at highest risk for complications is of paramount importance. The integration of our MDASI-PeriOp-BLC module into routine perioperative BLC care represents an opportunity to improve patient-centered care by allowing the assessment of patient-identified symptoms throughout the course of BLC treatment.

The management of BLC is a clinical challenge. BLC is more common among the elderly and rare in patients below the age of 50 years [45], which is also the case for this study sample (average 67 years old). Thus, comorbid conditions require special attention in treatment planning [4,46]. Furthermore, even with treatment advances, BLC remains among the five leading causes of cancer death among older men [37]. Radical cystectomy can result in bowel toxicity [47], urinary alterations, and sexual dysfunction [21,47]. Also, the treatment may negatively impact daily activities, social function [13,14], and body image [15].

The MDASI-PeriOp-BLC reflects the burden of the most critical BLC-associated symptoms during the perioperative period and their impact on daily activities. In this study cohort, bowel toxicity (constipation), urinary alterations (blood in urine, frequent urination, stomal problems, and leaking urine), and changes in sexual function were the most severely scored BLC module items, while fatigue was the most severe reported systemic symptom from MDASI-core. Increased symptom burdens of pain, poor appetite, and dry mouth negatively impacted daily activities, while the disease- and treatment-specific items of blood in urine, burning with urination, and stomal problems significantly impacted general activity. MDASI-PeriOp-BLC items could successfully differentiate patients according to time-sensitive symptoms (ex. pre- or postoperative), as well as with respect to ECOG performance status. Furthermore, QOL subscales were significantly correlated with the MDASI-PeriOp-BLC items, which indicates the sensitivity of the module items for determining QOL status.

In terms of comparing the performance of the MDASI-PeriOp-BLC to the existing PRO modules for BLC patients, significant correlations have been observed between MDASI-PeriOp-BLC subscales and the corresponding scores of FACT-Additional BL and EORTC-BLM30 items regarding urinary symptoms and problems, urostomy problems, future perspective, abdominal bloating and flatulence, and body image items. The BLC module items were not significantly associated with the item related to changes in sexual function of the EORTC-BLM30, which could be explained by the low response rate to this specific item.

The eight BLC-specific items of the MDASI-PeriOp-BLC that were generated from this study provide a brief PRO tool focused on BLC perioperative care. However, these items overlap with most of the items of the EORTC QLQ-BLM30, Functional Assessment of Cancer Therapy-Bladder (FACT-BL), and Vanderbilt Cystectomy Index of the Functional Assessment of Cancer Therapy (FACT-VCI) [48], as well as cover the domains of the Bladder Cancer Index (BCI) [49]. Yet, the EORTC QLQ-BLM30 is a 30-item instrument and is intended for use as a supplementary module to the EORTC QLQ-C30 [50]; BCI is a 36-item instrument, while the FACT-VCI is a 45-item instrument, and the FACT-Bl is a 39-item instrument [51].

This study carries several limitations. First, there are inherent limitations to a single-institution cohort. Additionally, the present study population is composed primarily of elderly non-Hispanic white males, and thus the results may not be applicable to other patient populations. Sexual function can be significantly impacted by BLC and its treatment—in the present study there was a low response rate to questions assessing sexual function, and respondents were primarily men—thus better assessment of perioperative sexual function changes may be best studied in a multi-institutional setting comprising a more diverse patient population. Furthermore, for successful PRO implementation in real-world patient care, there is a need for symptom research to better define thresholds of severity for each symptom and identify the most clinically meaningful values indicative of high-risk individuals necessitating monitoring and rapid intervention when warranted.

## 5. Conclusions

Through a combination of qualitative interviews, expert panel discussions, and cognitive debriefing, patients’ perspectives and clinicians’ points of view were utilized to develop and validate the MDASI-PeriOp-BLC module. In accordance with FDA guidelines for PRO measures, this study demonstrated the satisfactory psychometric properties (i.e., reliability and construct validity) of the MDASI-PeriOp-BLC as a disease- and treatment-specific PRO assessment instrument for BLC patients during the perioperative period. The MDASI-PeriOp-BLC was found to be brief, easy to use, and understandable.

## Figures and Tables

**Table 1 cancers-14-03896-t001:** Patient and disease characteristics (N = 150).

Characteristic	Mean	SD	Minimum	Median	Maximum
Age, years	67.7	9.66	39.71	68.49	89.43
Education level, years	14.7	2.13	7	16	17
Length of stay, days	7.67	6.29	3	6	73
	N			Percentage	
Sex					
Female	29			19.33	
Male	121			80.67	
Ethnicity					
Hispanic or Latino	20			13.33	
Not Hispanic or Latino	130			86.67	
Race					
Asian	1			0.67	
Black or African American	6			4.00	
White	141			94.00	
Other	2			1.33	
Marital Status					
Married/partnered	119			79.33	
Divorced	8			5.33	
Widowed	10			6.67	
Single, living with another adult	1			0.67	
Single, living alone	12			8.00	
ECOG performance status					
0–1 (good)	87			58.00	
2–4 (poor)	63			42.00	
Clinical stage					
0a	9			6.00	
0is	12			8.00	
I	51			34.00	
II	58			38.67	
III	12			8.00	
IV	7			4.67	
Unknown	1			0.67	
ERAS patient					
Yes	132			88.00	
No	18			12.00	
Surgical procedure					
Conventional urostomy (with stoma)	118			78.67	
Continent urinary reservoir (with stoma)	4			2.67	
Neobladder (without stoma)	28			18.67	
Recurrence					
Yes	37			25.17	
No	110			74.83	
Prior treatment					
Yes	118			78.67	
No	32			21.33	
Concurrent disease per Charlson Comorbidity Index					
None	108			72.00	
Present	42			28.00	

Abbreviations: SD, standard deviation; ECOG, Eastern Cooperative Oncology Group; ERAS, enhanced recovery after surgery.

**Table 2 cancers-14-03896-t002:** Internal consistency of MDASI-PeriOp-BLC items according to Cronbach’s alpha. (**A**) Core and BLC module items; (**B**) Interference items.

(A)		
	Overall α = 0.892	
Variable	No. of Patients	α if Item Deleted
Pain	104	0.881
Fatigue	104	0.882
Nausea	104	0.887
Sleep	104	0.879
Distress	104	0.883
Shortness of breath	104	0.888
Memory	104	0.887
Appetite	104	0.881
Drowsiness	104	0.879
Dry mouth	104	0.885
Sadness	104	0.886
Vomiting	104	0.890
Numbness	104	0.893
Blood in the urine	104	0.890
Frequent urination	104	0.892
Leaking urine	104	0.890
Burning with urination	104	0.891
Urinary urgency	104	0.891
Constipation	104	0.890
Changes in sexual function	104	0.898
Stomal problems	104	0.887
**(B)**		
	**Overall α = 0.899**	
**Variable**	**No. of Patients**	**α if Item Deleted**
General activity	145	0.868
Mood changes	145	0.876
Work	145	0.894
Relations	145	0.888
Walking	145	0.876
Enjoyment of life	145	0.885

Abbreviations: MDASI, MD Anderson Symptom Inventory; BLC, bladder cancer.

**Table 3 cancers-14-03896-t003:** Known-group validity of MDASI-PeriOp-BLC severity: (**A**) by performance status and (**B**) symptom severity at different time points (N = 150).

(A)								
Subscale	ECOG Score	N	Mean	SD	Lower 95% Confidence Limit	Upper 95% Confidence Limit	*p* Value	Cohen
MDASI core	Good (0–1)	87	1.53	1.22	1.27	1.79	<0.0001	−1.22
	Poor (2–4)	63	3.36	1.74	2.93	3.8		
MDASI interference	Good (0–1)	87	2.23	2.13	1.78	2.69	<0.0001	−0.94
	Poor (2–4)	62	4.58	2.8	3.87	5.29		
Physical MDASI (WAW)	Good (0–1)	87	2.66	2.61	2.11	3.22	<0.0001	−0.95
	Poor (2–4)	62	5.38	3.09	4.6	6.17		
Affective MDASI (REM)	Good (0–1)	87	1.8	2.02	1.37	2.23	<0.0001	−0.78
	Poor (2–4)	62	3.79	2.97	3.04	4.54		
BLC module items	Good (0–1)	85	1.21	1.31	0.93	1.5	0.3295	−0.21
	Poor (2–4)	60	1.53	1.68	1.1	1.97		
MDASI symptom severity	Good (0–1)	87	1.42	1.07	1.19	1.65	<0.0001	−0.97
	Poor (2–4)	63	2.67	1.47	2.3	3.04		
**(B)**								
**Variable**	**MDASI Conducted Time**	**N**	**Mean**	**SD**	**Lower 95% Confidence Limit**	**Upper 95% Confidence Limit**	** *p* ** **Value**	
MDASI core	pre-surgery	30	1.54	1.34	1.04	2.04	<0.0001	
	post-surgery < 5 days	52	3.29	1.58	2.85	3.72		
	post-surgery ≥ 5 days	68	1.88	1.65	1.49	2.28		
MDASI interference	pre-surgery	30	1.91	1.99	1.17	2.65	<0.0001	
	post-surgery < 5 days	51	4.47	2.78	3.69	5.26		
	post-surgery ≥ 5 days	68	2.84	2.52	2.22	3.45		
Physical MDASI (WAW)	pre-surgery	30	1.91	2.34	1.04	2.78	<0.0001	
	post-surgery < 5 days	51	5.35	3.09	4.48	6.22		
	post-surgery ≥ 5 days	68	3.46	2.9	2.76	4.16		
Affective MDASI (REM)	pre-surgery	30	1.91	2.12	1.12	2.7	0.0107	
	post-surgery < 5 days	51	3.61	2.99	2.77	4.45		
	post-surgery ≥ 5 days	68	2.21	2.37	1.64	2.78		
BLC module items	pre-surgery	30	1.78	1.61	1.18	2.38	0.0103	
	post-surgery < 5 days	49	1.64	1.75	1.14	2.15		
	post-surgery ≥ 5 days	66	0.93	1.03	0.67	1.18		
MDASI symptom severity	pre-surgery	30	1.63	1.28	1.15	2.11	<0.0001	
	post-surgery < 5 days	52	2.67	1.37	2.28	3.05		
	post-surgery ≥ 5 days	68	1.53	1.25	1.23	1.83		

Abbreviations: MDASI-PeriOp-BLC, MD Anderson Symptom Inventory Bladder Cancer Perioperative Module; REM, composite score of relations with other people, enjoyment of life, and mood, representing the MDASI’s mental health or social functioning domains; SD, standard deviation; WAW, composite score of work, activity, and walking, representing the MDASI’s physical functioning domain.

**Table 4 cancers-14-03896-t004:** Convergent validity of MDASI subscales using other validated instruments.

Instrument		MDASI Symptom Severity	MDASI Core	MDASI Module	MDASI Interference	Physical MDASI(WAW)	Affective MDASI(REM)
SIQOL	*r*	−0.398	−0.388	−0.214	−0.484	−0.427	−0.512
	*p*	<0.0001	<0.0001	0.0116	<0.0001	<0.0001	<0.0001
	No. of observations	141	140	138	139	139	139
FACT-Bl	*r*	−0.411	−0.365	−0.358	−0.449	−0.410	−0.442
	*p*	<0.0001	<0.0001	<0.0001	<0.0001	<0.0001	<0.0001
	No. of observations	138	137	135	136	136	136
EORTC-BLM30							
Urinary symptoms and problems	*r*	0.214	0.116	0.493	0.127	0.029	0.251
	*p*	0.1522	0.4477	0.0005	0.4117	0.8507	0.0998
	No. of observations	46	45	46	44	44	44
Urostomy problems	*r*	0.278	0.217	0.340	0.166	0.162	0.140
	*p*	0.0054	0.0322	0.0007	0.1048	0.1126	0.1726
	No. of observations	99	98	96	97	97	97
Catheter use problem	*r*	0.105	0.091	0.216	0.018	0.025	0.089
	*p*	0.5363	0.5919	0.2055	0.916	0.8854	0.6067
	No. of observations	37	37	36	36	36	36
Future perspective	*r*	0.292	0.268	0.272	0.352	0.250	0.455
	*p*	0.0006	0.0018	0.0017	<0.0001	0.0038	<0.0001
	No. of observations	134	133	131	132	132	132
Abdominal bloating and flatulence	*r*	0.450	0.460	0.267	0.481	0.442	0.463
	*p*	<0.0001	<0.0001	0.002	<0.0001	<0.0001	<0.0001
	No. of observations	134	133	131	132	132	132
Body Image	*r*	0.212	0.460	0.258	0.481	0.442	0.463
	*p*	0.0141	<0.0001	0.0031	<0.0001	<0.0001	<0.0001
	No. of observations	133	133	130	132	132	132
Sexual functioning	*r*	−0.034	−0.054	0.058	−0.002	0.030	−0.031
	*p*	0.7013	0.5416	0.5106	0.9839	0.7316	0.7244
	No. of observations	133	132	131	131	131	131

Abbreviations: MDASI, MD Anderson Symptom Inventory Module; REM, composite score of relations with other people, enjoyment of life, and mood, representing the MDASI’s mental health or social functioning domains; WAW, composite score of work, activity, and walking, representing the MDASI’s physical functioning domain; EORTC-BLM30, Organization for Research and Treatment of Cancer Quality of Life Core Questionnaire–muscle-invasive bladder cancer; SIQOL, single-item quality of life; FACT-Bl, Functional Assessment of Cancer Therapy–Bladder Cancer–Additional Items.

**Table 5 cancers-14-03896-t005:** MDASI item severity at pre-surgery and post-surgery time points.

Item	Pre-Surgery						Post-Surgery ≤ 4 Days						Post-Surgery > 4 Days						*p*
	N	Mean	SD	Med	Min	Max	N	Mean	SD	Med	Min	Max	N	Mean	SD	Med	Min	Max	
Pain	30	1.4	2.01	0	0	6	52	5.35	2.84	5	0	10	68	2.57	2.85	2	0	10	<0.0001
Fatigue	30	2.67	2.37	2	0	9	52	5.83	2.37	6	0	10	68	3.74	2.79	3	0	10	<0.0001
Nausea	30	0.73	2.05	0	0	10	51	2.41	3.22	1	0	10	68	1.22	2.32	0	0	9	0.0026
Sleeping disturbance	30	2.53	2.78	2	0	10	52	4.73	3.22	5	0	10	68	3.16	2.92	2	0	10	0.0041
Distress	30	2.5	2.69	2	0	9	52	2.38	2.52	2	0	9	68	1.66	2.41	1	0	10	0.0958
Shortness of breath	30	0.93	2.13	0	0	9	52	1.38	1.72	1	0	6	68	1.16	1.63	0	0	6	0.1333
Remember	30	1.3	1.47	1	0	5	52	2.25	2.05	2	0	7	68	1.1	1.79	0	0	8	0.0018
Poor Appetite	30	0.9	1.6	0	0	6	52	4.63	2.98	5	0	10	68	2.34	2.89	1	0	10	<0.0001
Drowsiness	30	1.93	2.12	1	0	7	52	5.02	2.36	5	0	10	68	2.43	2.56	2	0	10	<0.0001
Dry mouth	30	1.07	1.87	0	0	7	52	4.87	3.41	5	0	10	68	2.1	3	1	0	10	<0.0001
Sadness	30	2.03	2.36	1	0	10	52	1.9	2.3	1	0	8	68	1.46	2.17	0	0	9	0.2476
Vomiting	30	0.37	1.83	0	0	10	52	0.92	2.19	0	0	10	68	0.32	1.03	0	0	5	0.0798
Numbness	30	1.67	2.5	0	0	9	52	1.06	1.73	0	0	7	68	1.22	2	0	0	8	0.8102
Blood in urine	30	1.43	2.92	0	0	10	52	2.94	3.03	2	0	10	65	0.37	1.04	0	0	5	<0.0001
Frequent urination	30	3.4	2.94	3	0	10	45	0.58	1.88	0	0	10	65	0.65	1.49	0	0	8	<0.0001
Leaking urine	28	1.57	2.53	1	0	10	47	1.15	2.61	0	0	10	64	0.75	1.56	0	0	7	0.101
Burning with urination	30	1	2.15	0	0	10	46	0.41	1.65	0	0	10	66	0.18	0.89	0	0	7	0.0014
Urinary urgency	30	2.33	2.84	1	0	10	46	0.5	1.8	0	0	10	64	0.31	1.07	0	0	6	<0.0001
Constipation	30	1.43	2.03	1	0	7	51	3.39	3.44	3	0	10	68	1.99	2.8	1	0	10	0.0385
Changes in sexual function	28	2.32	3.48	0	0	10	46	1.85	3.41	0	0	10	40	3.1	4.29	0	0	10	0.3599
Stomal problems							51	1.59	2.84	0	0	10	68	0.84	1.62	0	0	10	0.1251
Activity	30	2.2	2.68	1	0	9	51	6.43	3.49	8	0	10	68	3.78	3.12	3	0	10	<0.0001
Mood	30	2.1	2.37	2	0	9	51	3.86	3.59	3	0	10	68	2.34	2.57	2	0	10	0.0531
Work	30	2	2.45	1	0	8	48	5.19	4.6	6	0	10	68	4.03	3.71	3	0	10	0.0179
Relations	30	1.2	1.79	0	0	8	50	2.28	3.3	1	0	10	68	1.66	2.33	1	0	10	0.6688
Walking	30	1.53	2.39	0	0	8	51	4.45	3.05	5	0	10	68	2.57	2.88	2	0	10	<0.0001
Enjoyment of life	30	2.43	2.98	1	0	10	51	4.53	3.57	5	0	10	68	2.63	2.93	2	0	10	0.0039
Core items	30	1.54	1.34	1	0	6	52	3.29	1.58	3	0	7	68	1.88	1.65	2	0	8	<0.0001
Interference items	30	1.91	1.99	2	0	9	51	4.47	2.78	5	0	10	68	2.84	2.52	2	0	10	<0.0001
WAW	30	1.91	2.34	1	0	8	51	5.35	3.09	5	0	10	68	3.46	2.9	3	0	10	<0.0001
REM	30	1.91	2.12	1	0	9	51	3.61	2.99	3	0	10	68	2.21	2.37	2	0	10	0.0107
Module items	30	1.78	1.61	1	0	7	49	1.64	1.75	1	0	9	66	0.93	1.03	1	0	5	0.0103
Severity items	30	1.63	1.28	1	0	5	52	2.67	1.37	3	0	7	68	1.53	1.25	1	0	6	<0.0001

Abbreviations: MDASI, MD Anderson Symptom Inventory Module; REM, composite score of relations with other people, enjoyment of life, and mood, representing the MDASI’s mental health or social functioning domains; WAW, composite score of work, activity, and walking, representing the MDASI’s physical functioning domain; BLC, bladder cancer.

## Data Availability

Not applicable.

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
