# Peer review of "Validation and Application of MD Anderson Symptom Inventory Module for Patients with Bladder Cancer in the Perioperative Setting"

_cancers, 2022, doi:10.3390/cancers14163896_

Round 1

Reviewer 1 Report

I would like to congratulate the Authors for their excellent and original work on a specific theme which represents a topic of relevance for both the oncologic and urology community. Results and interpretation form novel Inventory module assessing symptoms for bladder cancer represents indeed an always important argument to further assess. Nevertheless, the article in its current form deserves further revision before being considered suitable for publication. First and more importantly, the methods of the article are too extensive and limit the readability form the audience. I would suggest organizing differently the paragraphs in order to cluster all the statistical information in only one single last paragraph. Moreover, the way the MD Anderson Symptom Inventory module has been built necessitate deeper explanation in this section.

Additionally, even if the article led to significant results and apply significance to the field of interest the background and the discussion of the document remain far from the readership which may be not familiar with the topic. For this reason, I would suggest stressing the argument regarding existing evidence for other GU malignancies (e.g., PCa) on the impact of sympthoms assessment and their impact on the psychological aspects and the quality of life. On this aspect I would suggest to expand the bacground by briefly references such experiences in order to expand the understibility of your later objectives (DOI: 10.1111/and.13157). I look forward to see the document updated before considerring for further assessment. 

Author Response

We are respectfully re-submitting the manuscript, “Validation and application of MD Anderson Symptom Inventory module for patients with bladder cancer in the perioperative setting” for consideration for publication in Cancers.

 We appreciate the reviewer's comments. We responded to all comments and made some edits to satisfy the reviewer's comments and suggestions.

Reviewer 1

I would like to congratulate the Authors for their excellent and original work on a specific theme which represents a topic of relevance for both the oncologic and urology community. Results and interpretation form novel Inventory module assessing symptoms for bladder cancer represents indeed an always important argument to further assess. Nevertheless, the article in its current form deserves further revision before being considered suitable for publication. First and more importantly, the methods of the article are too extensive and limit the readability form the audience. I would suggest organizing differently the paragraphs in order to cluster all the statistical information in only one single last paragraph. Moreover, the way the MD Anderson Symptom Inventory module has been built necessitate deeper explanation in this section.

We would like to thank the reviewe for such positive feedback. For the development and validation of patient-reported outcome assessment tools, we made an effort to follow FDA guidelines that include specific stages of analysis. For simplicity, we added a summary for the reader at the end of the methodology section. Yet, we still need to keep the detailed methodology, following the FDA guidelines. In response to the reviewer's suggestion, we added “In summary, we followed the following steps: 1) we conducted interviewers with the patients to get their experience with BLC and radical cystectomy; 2) the interviews were recorded, transcribed verbatim by professional transcriptionists and analyzed; 3) we shared the final symptom list with expert panel; 4) we performed debriefing process to assess ease of completion, comprehensibility, acceptability, redundancy, use of the scoring system, item clarification, and content domain confirmation of the new instrument; 5) we assessed the reliability of the tool; 6) we assessed the validity of the tool by comparing the tool with the existing tools; 7) we assessed the applicability of the tool by assessing the patient compliance, by calculating the percentage of patients complete each item”.

Additionally, even if the article led to significant results and apply significance to the field of interest the background and the discussion of the document remain far from the readership which may be not familiar with the topic. For this reason, I would suggest stressing the argument regarding existing evidence for other GU malignancies (e.g., PCa) on the impact of sympthoms assessment and their impact on the psychological aspects and the quality of life. On this aspect I would suggest to expand the bacground by briefly references such experiences in order to expand the understibility of your later objectives (DOI: 10.1111/and.13157). I look forward to see the document updated before considerring for further assessment. 

We appreciate the reviewer's suggestion. We added “Further, it has been proven that considering patient perspectives is an essential component in patient-centered care in genitourinary setting, in general. For example, the functional and psychological aspects may influence the treatment choice and potentially the outcomes in patient with prostate cancer” to the introduction.

Reviewer 2 Report

Title: Validation and application of MD Anderson Symptom Inventory module for patients with bladder cancer in the perioperative setting

This study proposed and validated a symptom inventory module, i.e., MDASI-PeriOp-BLC module that aims to assess bladder cancer patients’ outcome after radical cystectomy. To develop the model, the 13 items in MDASI module were included, and additional symptoms that were frequently mentioned by patients were added. Then, items that lack of sensitivity were dropped to finally determine the item set. Statistical analysis showed that the module had high reliability and was able to significantly differentiated the patients by performance status. Overall, the study is OK to be published on Cancers after fixing small issues:

1.     Can authors conduct a comparison experiment, to validate that the proposed module is supreme to existing modules, like the original MDASI module?

2.     There are some typos in the paper. Between Line 255 to 256 there are two errors (r-35839) and (r=-41193), I suppose they should be r = -0.35839 and r = -0.41193. Please fix them.

3.     The fonts in tables are different from each other, should make them consistent.

Author Response

We are respectfully re-submitting the manuscript, “Validation and application of MD Anderson Symptom Inventory module for patients with bladder cancer in the perioperative setting” for consideration for publication in Cancers.

 We appreciate the reviewer's comments. We responded to all comments and made some edits to satisfy the reviewer's comments and suggestions.

Reviewer 2

This study proposed and validated a symptom inventory module, i.e., MDASI-PeriOp-BLC module that aims to assess bladder cancer patients’ outcome after radical cystectomy. To develop the model, the 13 items in MDASI module were included, and additional symptoms that were frequently mentioned by patients were added. Then, items that lack of sensitivity were dropped to finally determine the item set. Statistical analysis showed that the module had high reliability and was able to significantly differentiated the patients by performance status. Overall, the study is OK to be published on Cancers after fixing small issues:

  1. Can authors conduct a comparison experiment, to validate that the proposed module is supreme to existing modules, like the original MDASI module?

We thank the reviewer for the comment and appreciate the chance to clarify this point. We actually included the 13 symptoms of the MDASI Core items for the initial content domain for the module, see line 195. The final module includes the MDASI-core items, module-specific items, and the MDASI-interference items.  To validate this new tool, we also used the existing PRO tools of EORTC-BLM30, SIQOL, and FACT-BL. The disease/treatment-specific symptoms of MDASI-PeriOp-BLC are unique PROs (such as urgency and stomal problems) that have never been reported for use in patient care postoperatively for the BLC population.

  1. There are some typos in the paper. Between Line 255 to 256 there are two errors (r-35839) and (r=-41193), I suppose they should be r = -0.35839and r = -0.41193. Please fix them.

Thank you for the reviewer's comment. We have fixed these two errors.

  1. The fonts in tables are different from each other, should make them consistent.

We thank the reviewer for the suggestion. Since we are using the journal template, we were limited in space. We checked and corrected to ensure font consistency in the proof.
